# A Review of Symmetric Silicon MEMS Gyroscope Mode-Matching Technologies

**DOI:** 10.3390/mi13081255

**Published:** 2022-08-04

**Authors:** Han Zhang, Chen Zhang, Jing Chen, Ang Li

**Affiliations:** 1Institute of Electronic Engineering and Optoelectronic Technology, Nanjing University of Science and Technology Zijin College, Nanjing 210023, China; 2College of Telecommunications & Information Engineering, Nanjing University of Posts and Telecommunications, Nanjing 210003, China

**Keywords:** symmetrical gyroscope, modal matching, structural tuning, electrostatic tuning

## Abstract

The symmetric MEMS gyroscope is a typical representative of inertial navigation sensors in recent years. It is different from the traditional mechanical rotor gyroscope in that it structurally discards the high-speed rotor and other moving parts to extend the service life and significantly improve accuracy. The highest accuracy is achieved when the ideal mode-matching state is realized. Due to the processing limitation, this index cannot be achieved, and we can only explore ways to approach this index continuously. This paper’s results of error suppression for the symmetric MEMS gyroscope are initially classified into three categories. The first category mainly introduces the processing structure and working mode of the symmetrical gyroscope. The second is mechanical tuning from the structure and the third is electrostatic tuning from the peripheral control circuit. Based on the listed results, the paper compares the two tuning modes and analyzes their advantages and disadvantages. The fourth category is the tuning means incorporating the emerging algorithm. On this basis, the elements of improvement for future high-precision symmetric MEMS gyroscopes are envisioned to provide a part of the theoretical reference for the future development direction of sensors in inertial navigation.

## 1. Introduction

Since the beginning of MEMS sensors in inertial guidance in the twentieth century, numerous high-precision and high-reliability sensors have been used. Many research scholars have launched extensive exploration for error suppression and performance improvement to explore the performance of the perfect MEMS gyroscope.

In the background to the rapid development of integrated circuits and silicon processing, Muller’s research team at the University of California, Berkeley, was the first to propose the concept of “micro-electromechanical systems” (MEMS) with intelligent structures based on silicon integration processes in the range of 1 μm to 1 mm. This ultra-miniature electrostatic motor has a simple structure and high no-load speed compared to the electromagnetic motor, suitable for the electrostatic drive of microsystem, so the application of electrostatic motor has been mainly focused on MEMS sensors in many countries since.

MEMS gyroscopes used to measure the angular velocity of the carrier are the most common types of inertial sensors, and consumer gyroscopes mostly use the tuning fork structure. However, because this gyroscope is extremely sensitive to the temperature coefficient, it is not destined to become a member of the high-precision sensor family. Dual mass, hemispherical, ring, and other symmetric structures of MEMS gyroscope can make the angular rate measurement insensitive to irrelevant variables, such as ambient temperature and common mode noise, thus improving the gyroscope reliability and anti-interference, widely used in the environments of high precision and high signal-to-noise ratio signal acquisition [1,2,3,4,5]. In the current public gyroscope test data, high precision indexes are done mainly by the hemispherical gyroscopes. Due to the complex processing process, two-dimensional axisymmetric gyroscopes are cost-effective choices [5,6]. The symmetric MEMS gyroscope meets the increasing demand for position services.

The main structure of an axisymmetric MEMS gyroscope generally consists of a resonator and an encapsulated housing. The resonator physically exists in two operating modes, and the gyroscope measures the attitude control information of the external carrier based on the energy transfer of the two modes. Supposing the removal of the drive mode is x and the removal of the detection mode is y, combined with the second-order equation of motion, when there is an external angular velocity input, then the motion of the two modes of the gyroscope can be described as [7,8,9,10,11]:(1)$$\frac{d^{2}x}{dt^{2}}+\frac{\omega_{x}}{Q_{x}}\frac{dx}{dt}+\omega_{x}^{2}x=\frac{F_{d}\sin\left(\omega_{d}t\right)}{m}\;$$
(2)$$\frac{d^{2}y}{dt^{2}}+\frac{\omega_{y}}{Q_{y}}\frac{dy}{dt}+\omega_{y}^{2}y=2\Omega_{z}\frac{dx}{dt}$$
where
m—displacement of mass block$\omega_{x},\;\omega_{y}$—resonant frequency of drive and detection modes$Q_{x},Q_{y}$—quality factor of two modes$\Omega_{z}$—external input angular velocity$\omega_{d}$—angular frequency of external input driving force

Solving the Equations (1) and (2), the stable displacements of the two modes are obtained as ($\omega_{d}=\omega_{x}$)
(3)$$x\left(t\right)=\frac{F_{d}/m}{\sqrt{\omega_{x}^{4}/Q_{x}^{2}}}\sin\left(\omega_{x}t+\varphi_{x}\right)\;$$
(4)$$y\left(t\right)=\frac{-2\Omega_{z}F_{d}Q_{x}}{m\omega_{x}}\cdot \frac{1}{\sqrt{{\left(\omega_{y}^{2}-\omega_{x}^{2}\right)}^{2}+\omega_{y}^{2}\omega_{x}^{2}/Q_{y}^{2}}}\sin\left(\omega_{x}t+\varphi_{y}\right)$$

The mechanical sensitivity, $S_{y}$ of the symmetrical gyroscope is:(5)$$S_{y}=\left|\frac{y\left(t\right)}{\Omega_{z}}\left|=\right|\frac{2F_{d}Q_{x}}{m\omega_{x}}\cdot \frac{1}{\sqrt{{\left(\omega_{y}^{2}-\omega_{x}^{2}\right)}^{2}+\omega_{y}^{2}\omega_{x}^{2}/Q_{y}^{2}}}\right|$$

According to Equation (5), the smaller Δω ($\omega_{y}-\omega_{x}$) is, the higher the mechanical sensitivity of the gyroscope.

When the resonant frequencies of the two working modes are equal, the displacement of the two modes is:(6)$$x_{z}\left(t\right)=\frac{Q_{x}F_{d}}{m\omega_{x}^{2}}\sin\left(\omega_{x}t+\varphi_{x}\right)\;$$
(7)$$y_{z}\left(t\right)=\frac{-2\Omega_{z}F_{d}Q_{x}Q_{y}}{m\omega_{x}^{3}}\cdot \sin\left(\omega_{x}t+\varphi_{y}\right)$$

When the gyro operates in force feedback mode, the condition of $\omega_{x}=\omega_{y}$ produces the highest sensitivity.
(8)$$S_{ymax}=\frac{2F_{d}Q_{x}Q_{y}}{m\omega_{x}^{3}}$$

When the gyro is operating in full-angle mode, the condition of $\omega_{x}=\omega_{y}$ minimizes the quadrature error. Therefore, the symmetry of the resonant oscillator’s two modes determines the sensor’s operating performance. At this time, the mechanical sensitivity $S_{y}$ reaches the maximum value. The higher the sensitivity, the higher the signal-to-noise ratio of the system. At the same time, the driving amplitude is also reduced. Due to the randomness of processing errors, the two modes of the resonator cannot be highly matched. At the same time, the motion parameters of the driving and detecting modes are coupled to produce orthogonal errors, which reduce the sensitivity and resolution of the gyroscope, deteriorate the zero-rate bias and stability, and significantly reduce the performance [7,10,12]. Therefore, it is necessary to propose practical mode-matching approaches.

The technical routes for error suppression in symmetric resonant gyro-based measurement systems can be summarized in two main ways. One way of mode-matching starts by improving the structure. This kind of method is usually aimed at adjusting the small range of error of a high precision gyroscope, generally based on structural modeling using unique processing methods to change a specific parameter of the resonant oscillator until it achieves the mode-matching. 

There are two primary technical means to achieve structural tuning. One is to change the equivalent mass by depositing or etching the resonator to change the frequency [13,14,15,16,17,18] according to the relationship between the resonator equivalent mass m and the mode frequency, $\omega_{y}$.
(9)$$\omega_{y}=\sqrt{k_{y}/m}\;$$

Another way to achieve mechanical tuning is to use new structures or materials [19,20,21]. This modification is highly innovative but risky, and few relevant results are available. In 2019, the researchers at Peking University designed a new gyroscope structure [22], macroscopically shaped like a “Christmas tree.” The primary material of the beam structure is < 100> monocrystalline silicon (SCS) instead of <111>. The design structure reduces the frequency difference and the processing cost of <100> SCS compared to <111>. This design is also a significant breakthrough in our symmetric MEMS gyroscope, using structural innovation to solve the modal mismatch problem.

The other technical route of mode-matching is to start from the peripheral control algorithm, which can also be called electrostatic tuning. The tuning method is based on the principle of the negative stiffness effect to decouple the two working modes (when the resonator has a coupling error). It can change the DC control voltage on the resonator electrode to adjust the mode-matching error to an acceptable range. This method originated at the University of Michigan in 1997 and students discovered that the principle of the filament annealing technique is also applicable to microstructures. The annealing technique focuses on placing semiconductor materials at a specific high temperature and injecting impurity ions, which can be used to restore the crystal structure [23]. This method of eliminating material defects was applied to the silicon-based material of the gyroscope. The result did change the resonant frequency of the polysilicon mechanical resonator and could improve the quality factor [24,25].

The technical means to achieve voltage tuning can be divided into three main categories. The difference lies in the physical quantity of the tuning reference. The whole symmetric MEMS gyroscope can be viewed as a second-order system. The degree of resonator mode mismatch affects the frequency response. We can use the amplitude [26,27,28,29,30] and phase [31,32] as the basis of modal matching. The third way of voltage tuning combines two physical quantities, amplitude, and phase, as a reference for mode-matching. This tuning can reduce the errors caused by chance factors and achieve higher accuracy. For symmetric MEMS gyroscopes with center frequencies up to 100 kHz, the frequency difference after tuning is more minor than 1 Hz [25,33,34].

Since the end of the twentieth century, many research institutes have been dedicated to exploring the accuracy breakthrough of symmetric MEMS gyroscope and there have been remarkable results. Previous reviews of symmetrical gyroscopes are generally based on structure, materials and working principles and there is a less in-depth analysis of error suppression technology. However, the accuracy of the symmetrical gyroscope is an important parameter to reflect its performance. To understand the method of improving the accuracy of the gyroscope, this paper classifies and summarizes some classical achievements. Based on the existing achievements, the direction of future development can be determined, and a theoretical basis for researchers in related fields provided. In the following paper, some representative results will be cited to illustrate this. The following content is mainly divided into four parts.

The first part introduces the basic types and main characteristics of symmetrical gyroscopes. The advantages and disadvantages of different operation mechanisms are presented on this basis.The second part introduces the structure-based mechanical tuning of symmetric MEMS gyroscope. This tuning method is subdivided into two categories, and the performance is compared and summarized.The third part introduces the voltage tuning based on the peripheral control circuit. This tuning is subdivided into three types, and the performance is compared and summarized. The advantages and disadvantages of mechanical tuning and electrical tuning are compared. Based on the current situation, the critical optimization direction of electrical tuning is proposed.The fourth part introduces the tuning method incorporating emerging algorithms. The fuzzy and neural network control are typical algorithms [35,36,37,38].The last part is based on the above status analysis to prospect the future development direction of the symmetric MEMS gyroscope.

## 2. Basic Features and Critical Parameters

Symmetric gyroscopes mainly include tuning fork vibration gyroscope, vibrating ring gyroscope (VRG) and hemispherical resonant gyroscope (HRG) [14]. They all belong to the Coriolis vibration gyroscope. This section mainly introduces the symmetrical gyroscope’s different structural materials and operating mechanisms.

### 2.1. Drive and Sense Method

Coriolis vibration gyroscopes have two crucial parts, the drive and sense parts. The general drive part adopts the electrostatic excitation method, and the sense part adopts the capacitive detection method [14,19,20,21,39,40].

There are three primary forms of electrostatic actuation: angular vibration, line vibration and standing wave vibration. The capacitive detection method has two primary structure forms: angular vibration and line vibration. Therefore, there are various combinations of drive-sense mode, as needed.

The tuning fork vibratory gyroscope is the ancestor of the vibratory gyroscope. After years of careful research, J. Lyman and R.E. Barnaby of the Sperry company first disclosed their discovery, the Sperry tuning fork vibratory gyro (gyrotron), in the United States in 1953. The design is the first vibratory gyroscope in the world. Since 1993, the Charles Stark Draper Laboratory (CSDL) (Cambridge, MA, USA) and Rockwell International Corp. (Milwaukee, WI, USA) researchers have been working to commercialize CSDL’s silicon tuning fork gyro in the automotive industry [41]. Tuning fork gyroscopes generally use linear and angular vibration methods.

Angular vibration was a more common drive method in the early days of symmetrical gyroscopes. In 1988, the research team in the Draper Laboratory developed an inner and outer dual-frame silicon miniature angular vibration gyroscope [42]. The outer frame is used for driving and the inner frame is used for sensitive external angular velocity input. This type of vibration generally requires a structure with two frames, an inner and an outer frame. The outer frame of the instrument, together with the two internal frames and the mass block, rotates at high frequencies and with small amplitude under electrostatic actuation.

In 1993, the Draper Laboratory’s researchers improved the dual-frame gyroscope with a line vibration drive and angular vibration sensitivity. The performance index has been dramatically improved. The adequate size is 1 mm and the drive amplitude is six μm. The Q values of the drive and sense axes are 40,000 and 5000, respectively [26,27,43]. The linear vibration drive relies mainly on the comb drive. When an AC signal is applied to the fixed pole plate of the driver, the moving pole plate is displaced. The gyroscopic effect occurs when the external angular velocity is used [28]. Compared with the angular vibration drive, the line vibration drive has the following advantages:Angular vibration requires a double frame structure. This gyroscope fabrication process is more complex, while the processing of comb-driven gyroscope is relatively simple.The comb driver uses the surface processing method. This method is compatible with the line integration process.Due to the second point of compatibility, the integration of the gyroscope detection chip is also improved.

Therefore, line vibration is the most commonly used excitation and detection method. In 2021, Nanjing University of Science and Technology designed a decoupled bilinear vibration gyroscope. At a vibration level of 7.0 g, the vibration variation is <10°/*h* and the zero bias stability is close to 10°/*h* over the whole temperature range. This test level is only one order of magnitude away from the tactical level (1°/*h*) [29].

The other two symmetrical gyroscopes are the VRG and HRG. Their working principle is the same. Therefore, the excitation-detection principles of both are the same. The VRG and HRG are driven and sensed using standing wave vibrations. As a vibrating structure rotates, the standing wave also rotates due to inertial forces. The angular rotation rate is proportional to the angular rotation rate of the external carrier, and the proportion is fixed [30,44,45,46]. This phenomenon is called the “Bryan effect”.

Although the VRG and HRG work on the same principle, the advantages and disadvantages differ significantly. They have significant differences in preparation materials and processing methods. The VRG is a surface processing method and the HRG is a body processing method. The accuracy of hemispherical gyroscopes is comparable to that of laser gyroscopes, mainly used in aerospace. The United States was the first to start researching the hemispherical resonant gyroscope (HRG). In addition, Russia’s research on hemispheric gyroscopes has been very fruitful. In 1993, an HRG with a diameter of 100 mm and 50 mm was developed by the Russian Ramin Design Bureau (Saint Petersburg, Russia). The performance indexes are a random drift of 0.001~0.005°/*h*, a random wander of 0.005–0.01°/√h and a lifetime of up to 20 years [31,47]. However, hemispherical gyroscopes are smaller and simpler systems compared to optical gyroscopes.

The machining of the HRG resonator is very complex, so the ring gyroscope with surface machining was born. The VRG is divided into single-ring and multi-ring (disc). Flexible beams connect multi-ring structures and have electrodes embedded between the rings, increasing the drive and detection capacitance. In 2000, F. Ayazi and K. Najafi of the University of Michigan (Ann Arbor, MI, USA) designed a polycrystalline silicon ring gyroscope with 80 μm thickness, 1.1 mm diameter, and 20:1 aspect ratio. The gyroscope was tested to have a resolution better than 1°/s in a 1 Hz bandwidth and this result is one of the more classic designs in the early stages of VRG development [48].

### 2.2. Materials

In the past, quartz crystal was the primary material for making sensors. Quartz material has an excellent piezoelectric effect and temperature stability. Importantly, even in a non-vacuum environment, quartz can achieve a high Q value, which helps to improve the sensitivity and SNR of the gyroscope. With the development of microsystem technology, researchers have higher and higher requirements for gyroscope integration.

However, quartz is hard and brittle, which is unsuitable for micromachining [32,49,50,51,52,53]. Therefore, people began to use silicon instead of quartz crystal. The Youngs modulus of silicon is directional, which will cause more significant mechanical coupling and zero drift between driving mode and vibration mode [54,55,56].

Since the 1980s, fused quartz has been utilized again. Fused quartz is isotropic and has a minimal expansion coefficient. Fused quartz has no anisotropy and has a very low expansion coefficient and impurity level. In this way, the thermoelastic loss and volume internal friction are reduced. At the same time, the quality factor of the structure is increased. In recent years, hemispherical gyroscopes with high precision and high Q value have been prepared by fused quartz [57,58]. In 2021, the research team at the National University of Defense Technology introduced a micro hemispherical resonator with teeth-like tines along the edge. The *n* = 2 wineglass modes operate at 4.3 kHz, achieving a quality factor of 1.18 million [59]. The same year, the research team at Wuhan University presented a high-performance piezoelectric ring resonator with an ultra-low relative frequency split. It achieves an ultra-low relative frequency split of 11.2 ppm at its resonant frequency of 456.68 kHz. The measured Q values are between 11,391 and 15,324, with an average of 13,296 [60]—two orders of magnitude difference in quality factors between VRG and HRG.

### 2.3. Operating Mechanism

There are two operating mechanisms of tuning fork gyroscope: open-loop detection and closed-loop detection, collectively referred to as rate mode or amplitude modulation (AM) mode. In open-loop detection, the detection displacement amplitude of gyro output is proportional to the input angular velocity and the demodulated voltage reflects the angular velocity information.

In Figure 1, $G_{y}\left(s\right)$ is the transfer function of the gyroscope detection mode; *K* is the forward path gain; $F_{lpf}\left(s\right)$ is the low-pass filter transfer function in demodulation; $V_{ref}\sin\left(\omega_{d}t+\varphi_{0}\right)$ is the modulating signal; $V_{open}\left(t\right)$ is the output signal, which can reflect the angular velocity. The open-loop control system has a simple model and high loop gain. However, the system stability is poor and few scenarios are currently applied. In the closed-loop detection (force balance detection), a force feedback detection electrode is added to the gyro detection mode. The feedback force of the closed loop counteracts the Coriolis force. Feedback force reflects the magnitude of angular velocity. It is characterized by excitation mode constant amplitude vibration, and the detection mode displacement is kept at zero. In the force balance mode, the gyroscope scale factor is independent of the forward path gain and is only related to the feedback gain. This operating mechanism dramatically improves the scale-factor stability. In addition, the response is usually linear because the detected modal motion is always near the zero position. In the various electrical tuning methods mentioned later, as described in [61,62,63,64], gyroscopes are based on a force feedback operating mechanism [65,66].

In addition to the AM mode, the operating mechanism of the VRG and HRG have a whole-angle (WA) mode, also called the rate-integrating mode. The WA mode is one of the most typical applications of the Bryan effect. When there is an external angular velocity, the standing wave of the resonator and the electrode rotate with the carrier. However, the rotation of the standing wave always lags behind the course of the base by a fixed ratio. This fixed ratio is called the feed factor or Bryan factor. The carrier’s motion can be deduced by combining the standing wave rotation angle and the feed factor.

Since the WA mode directly outputs the angular information of the carrier, it can avoid the drift introduced by integrating the angular rate gyroscope, and it has the advantages of large bandwidth, high dynamics, and high linearity. However, the standing wave axis is inferred using the proportional relationship between the 0° and 45° signals. It is easier to introduce quantization errors in this process, resulting in lower measurement resolution in the WA mode. The researchers at the University of California presented a generalized electronic feedback method for compensating resonator anisodamping and anisoelasticity using the WA operation in a new way. The operation overcomes the precession angle-dependent bias error and minimum rate threshold. This method has decreased the angle-dependent bias error by 30, resulting in a minimum rate threshold of 3.5 DPS. The RIG’s output noise is also evaluated, demonstrating an ARW of 11 mdps/√Hz, similar to rate gyro operation at the same amplitude [67].

## 3. Mode-Matching Technology Based on Mechanical Tuning

Symmetrical gyroscopes are widely used in inertial navigation systems and it is necessary to understand their error suppression techniques. Therefore, the following three sections make a basic overview based on this content. In this paper, error suppression techniques are mainly divided into two categories. One is mechanical tuning and the other is electrical tuning. Mechanical tuning technology starts from the structure to adjust the gyroscope’s small range of error, applicable to the processing process before molding. The general object of tuning is mainly to test instruments with high requirements for accuracy, and the process is generally theoretical simulation followed by the actual processing. There are two types of mechanical tuning. One is to proceed with changing the resonator mass, and the other is to proceed with changing the structure and material. Here are the details of each.

### 3.1. Mode-Matching Approach with Fine-Tuning of Structural Parameters

Since the gyroscope can be viewed as a classical second-order system, there is a relationship between its modal frequency and equivalent mass (Equation (7)). One of the means of mechanical tuning is to change the mass and stiffness of the symmetric gyroscope and further change the mode frequency, such as mass deposition, resonator coating, laser etching, etc. In 2005, researchers at the Georgia Institute of Technology (Atlanta, GA, USA) proposed a method for fine-tuning the modal frequency of MEMS resonators by adding gold-plated single crystal silicon to the upper surface of the resonator. The resonator’s increased mass where the silicon is deposited changes the equivalent stiffness and modal frequency. This tuning method achieves a high level of sensitivity but requires precise knowledge of the resonator mode nodes [15].

The above method is a purely mechanical tuning, and the controllability needs improvement. In 2009, the researchers at UCLA combined electromagnetic driving and capacitive sensing to form a control system. A disk-shaped NdFeB magnet is placed on a resonator to simulate reversible mass perturbations, as shown in Figure 2. The electromagnetic actuator generates a varying current through its solenoid valve to apply a radial force to the resonator. The capacitor is a brass disk parallel to the resonator’s outermost edge—the capacitance between the resonator and the brass disk changes as the resonator vibrates. Test data showed that the modal frequency changes at about 0.2 Hz per unit mass [16].

Due to the presence of electromagnetic devices, the object under test must be considered less susceptible to interference from the electromagnetic environment. Care must be taken to also isolate the high-frequency current from which the gyroscope drives the modal. The research team proposed improvements to the above method. They studied the tuning scheme with mass deposition in the outer layer of the gyroscope spokes, as shown in Figure 3. 

In the Figure 3, the two electromagnetic actuators are labeled D1 and D2, and the two capacitive pick–offs that detect planar deflection of the resonator are located at S1 and S2. The deposition mass and position could be quantified according to the tuning model. After tuning, the modal frequency difference was less than 0.1 Hz for a ring gyro with a resonant frequency of 14 kHz when the frequency mismatch exceeded 30 Hz, as tested by the prototype [17].

In 2021, the National University of Defense Technology (Changsha, China) researchers reported a novel ring MEMS resonator and a novel method of mechanical frequency tuning. The most prominent characteristic of the resonator is that 16 raised mass blocks are increased uniformly in the circumferential positions of the ring. However, the stiffness of the resonator is almost unchanged, which shows that the scheme can realize the decoupling of mass and stiffness. The frequency difference has a linear relationship with the changed quality [68].

The above “mass addition” tuning method increases the equivalent mass of the gyroscope, resulting in increased system power consumption. In 2001, the researchers at the University of Nottingham (Nottingham, UK) introduced the concept of “equivalent defect mass” for the first time, which was also the first time that the idea of “structural mass subtraction” was used. According to the Rayleigh–Ritz method, the researchers analyzed the effect of resonator mass imbalance on the modal error. They proposed the process of modal trimming by eliminating these mass imbalance errors at specific locations of the resonator. Still, this method only applies to trimming the edges of single and two pairs of oscillators [18].

Gallacher’s research team at Newcastle University (Newcastle, UK) investigated the use of laser ablation to achieve modal frequency trimming of non-ideal ring structures in 2003. Researchers theoretically analyzed how the ablation results change the intrinsic frequencies of both in-plane and out-of-plane deflections, including the effect of mechanical stiffness and mass reduction due to ablation on the frequency shift [69]. The team performed an experimental verification. The resonator laser ablation process is shown in Figure 4.

In the early stage, this kind of method was applied to the modification scene with high accuracy requirements. All of the above are modal matching using mass increase method and mass decrease method. The comparison of experimental data is shown in Table 1.

### 3.2. New Materials or Structures for Mode-Matching Approach

The other mechanical tuning is to retain the energy conversion working principle of the symmetric gyroscope but completely abandon the past’s more common gyroscope processing structure and materials. The design changes are more thorough than the previous one.

Silicon and quartz are the primary preparation material. However, there are also research teams using new materials and structures to reduce the disadvantages of traditional processing methods affecting the accuracy of the gyroscope in improving sensor performance. When considering the use of new materials and processes, we should not limit ourselves to the purpose of accuracy improvement but also consider the cost, power consumption, and safety compared with the original solution. Therefore, the index test is also multifaceted.

This tuning approach was gradually developed in the early 21st century, with a relatively short research history. In 2001, a research team at the University of California, Berkeley, integrated a mechanical beam structure composed of a thermistor into a comb micro-resonator for frequency tuning based on the local thermal stress effect. After heating, a resonator with a center frequency around 31 kHz was measured to change to 6.5%. Subsequently, hundreds of millions of frequency tunings were performed, and no visible material breakage was found on the resonator [61]. According to the experimental results, the research team established an electro-thermal dynamic model to improve the theoretical study. The simulation curve is shown in Figure 5.

While this tuning method had outstanding reliability, power consumption, and sensitivity advantages, replacing the mechanical beam material with a thermistor would increase the temperature drift of the system. This result reduced the system temperature stability, and it was not suitable as a measurement and control device for variable temperature environments.

In 2017, a new honeycomb disc resonant gyro was developed by researchers at the National University of Defense Technology (shown in Figure 6), whose topology consists of several interlaced hexagonal cells. The design had the advantages of high space utilization and excellent mechanical properties. Test results showed that the inherent frequency of the resonator was about 17,140 Hz, and the frequency cracking could be reduced to less than 0.1 Hz after electrostatic tuning [22].

In 2019, a new spider-web type disc resonant gyroscope (CDRG) was designed by the research team at the Soochow University, as shown in Figure 7. Its linear structure reduced the manufacturing defects dramatically compared with the DRG (RDRG). The performance tests of CDRG and RDRG were compared, and the results showed that the minimum frequency split of CDRG was 10.8 times smaller than the relative frequency split of RDRG built side-by-side on the same wafer. The maximum vibration displacement, scale factor, angular random wander, and bias instability were improved by 62.8%, 112%, 700%, and 314%, respectively [23].

Gyroscopes are also used in car navigation systems. To ensure impact resistance and high precision, Murata Manufacturing (Kyoto, Japan) and Kanazawa Murata Manufacturing (Hakusan, Japan) reported a precise MEMS gyroscope with a ladder structure in 2007. The fabricated gyroscope sensor module showed low acceleration sensitivity and stability [70].

The Q value of the above design was too low, reducing the accuracy and not meeting the autonomous driving specification. After that, the Department of Device R&D co-operated with the University of Michigan, composing a ladder structure, as shown in Figure 8. Capable of suppressing energy dissipation, the ladder structure allowed the Q-factors of drive and sense modes to reach 120,000, an excellent value as a Si-tuning fork gyroscope. The mode-matched high Q-factor ladder gyroscope showed a random angle walk (ARW) of 0.020°/√h and a bias instability (BI) of 0.20°/*h* [71].

The tuning process of the ring and hemisphere gyroscopes requires two steps to complete. The first step must be to decouple the two operating modes, and then the negative stiffness effect is used to change the modal frequencies until they match. Therefore, mechanical tuning can be done by employing some materials with less modal coupling or design. The researchers at Politecnico di Torino (Torino, Italy) presented a new, structurally and thermally stable design of a resonant mode-matched electrostatic z-axis MEMS gyroscope. The novelty of the proposed MEMS gyroscope design lay in the implementation of two separate masses for the drive and sense axis, using a unique mechanical spring configuration that allowed minimizing the cross-axis coupling between the drive and sense modes [72].

Considering the mode mismatch caused by temperature change, a comb-driven electrostatic tuning electrode is added to the gyroscope.

New structural design is an early mode matching method. Error adjustment accuracy is generally not high. The modal matching results of the above different design methods are shown in Table 2.

### 3.3. Summary

The most significant advantage of mechanical tuning is that it can be done once and for all. However, the preliminary work for the retuning, such as error modeling of the structure, has to be sufficient. For example, the deviation of the drilling rig angle during the processing will cause the stiffness and frequency of the gyroscope to be asymmetrical. The researchers at the Nanjing University of Science and Technology proposed a novel method for analyzing the effect of rig error on the angle. The process overcame the defects of the traditional methods and an explicit solution to the RIG angle error was obtained [73]. The need to know enough about the structure and processing of gyroscopes was the key to understanding the main factors affecting the accuracy and establishing the structural model scientifically.

The mechanical tuning method above, based on structural parameters, is generally used in the environment of high-precision measurement requirements. However, this tuning process is usually more expensive, as the structure of the incremental method requires the use of precious metals and the processing of new designs and raw materials lacks a pipeline. Changing the design after finalization is difficult, which is not helpful for the application scenarios requiring secondary modulation. Therefore, various algorithms of electrical modulation have gradually appeared.

## 4. Electrostatic Stiffness-Based Mode-Matching Technology

The mode-matching technique based on electrostatic stiffness is the second classical way to suppress the frequency error of a symmetric gyroscope, first, by determining the degree of mismatch, then by picking the tuning voltage and reducing the frequency difference through P-I-D negative feedback [24,25,33,34]. The core of this type of technology lies in determining the tuning voltage given the amplitude-frequency response and phase-frequency response of the frequency response of the gyroscope control system. Therefore, the determination of the tuning voltage relies on two references; one is the amplitude and the other is the phase.

Voltage tuning generally does not require structural modeling and its difficulty lies in the lack of compatibility of various algorithms in the face of different measurement and control systems. Typical research findings of tuning using two references are summarized below.

### 4.1. Amplitude Reference Determines the Tuning Voltage

In 2008, researchers at the University of California, Berkeley, designed a gyroscope readout circuit with real-time online calibration. The block diagram of the scheme design is shown in Figure 9. The basic principle is to use two equal-amplitude signals applied to the gyroscope’s driving mode, one with frequency $f_{d}\;$+ $f_{c}$ and one with frequency $\;f_{d}\;$ − $f_{c}$ ($f_{d}$ was the magnitude of the drive frequency, $f_{c}<f_{d}$). Thus, we can obtain the mismatch information by observing the asymmetry degree of the two frequency signals of the detection mode.

The control algorithm could reduce the frequency difference to below 50 Hz, and the noise and power consumption was meager. The research team also integrated the algorithm circuit using a 0.35 μm CMOS process [74].

In 2009, a research team from Trento designed an adaptive controller for mode-matching errors. The controller’s primary function was to find the extremes of the system response in the mode-matching state. The schematic block diagram of the control algorithm is shown in Figure 10. The main advantage of the controller design was that it was highly operable and could be implemented using a few analog devices. The scheme was tested on the LISY300AL transverse pendulum micromechanical gyroscope from ST Microelectronics and the experiments showed that the modal frequency difference could be controlled to within 1 Hz [61].

The prerequisite for the tuning algorithm was that the frequency difference identification algorithm needed to be solidified in the circuit. Hence, the model must be supported by many existing acquisition data and complex algorithm calculations.

In 2012, researchers at the Middle East Technical University (Ankara, Turkey) obtained the mismatch information by injecting a perturbation signal into the orthogonal traffic channel. The researchers chose the tuning voltage by simulating the ratio of the in-phase and orthogonal components of the gyroscope response as a mismatch reference. The system block diagram and test results are shown in Figure 11. The noise was reduced by a factor of 6 and the thermomechanical noise was reduced to 0.4°/h/√Hz [62].

In 2018, the research team at the Southeast University (Dhaka, Bangladesh) reported a mode-matching algorithm based on quadrature modulation, which was also based on the technical means of the applied perturbation signal. However, they utilized a more intuitive demodulation amount to judge the modal mismatch. The researchers used a sinusoidal signal with a frequency slightly larger than that of the operating bandwidth of the gyroscope to the quadrature stiffness correction electrode, which constituted a modulated quadrature force. The quadrature force was demodulated and filtered to obtain the frequency difference information [63]. The system block diagram is shown in Figure 12.

Based on previous results, the group improved the above design by using a sinusoidal signal with a signal whose frequency was slightly larger than the gyroscope bandwidth. The modal match was determined based on detecting the symmetry of the modal response to that signal. The two metrics were improved by 3.25 and 4.49 times compared to the frequency detuned case [64]. In 2021, a research team from Nanjing University of Science and Technology used a similar control algorithm to act on a disk resonator gyroscope (DRG) operating in the *n* = 3 wineglass mode. It also has a noticeable tuning effect. The compensation method can effectively eliminate the tuning error and reduce the frequency split from ∼0.5 Hz to <0.01 Hz [75].

The above mode-matching methods require an additional disturbance signal (which can also be seen by combining the system block diagram). The degree of modal mismatch is then judged based on the frequency response. A disadvantage of such an operation is that it will disturb the operating state of the original system. It also means that the frequency difference information cannot be judged in real time. In 2021, the researchers at Southeast University proposed the mode-matching method. The method took advantage of the power symmetry of the noise and the Coriolis response within the upper sideband (USB) and the driving frequency’s lower sideband (LSB). During the matching control, the history of the estimated mode order was recorded to adaptively tune the controller gain, speeding up the tracking process and reducing the steady ripples. By the mode-matching, the scale factor of the gyroscope under test was boosted by ten times; the angle random walk was enhanced from 8.13°/*h*/sqrt(Hz) to 1.99°/*h*/sqrt(Hz) and the bias instability was improved from 2.09°/*h* to 1.12°/*h* [76].

The above control models are modal matching based on amplitude. The experimental phenomenon of this kind of method is intuitive. It is also a classic mode matching method. The comparison of experimental data is shown in Table 3.

### 4.2. Phase Reference Determines the Tuning Voltage

Based on the characteristics of the second-order system of the gyroscope, its phase-frequency response can also be used as a reference for modal matching. When the symmetric MEMS gyroscope is in the mismatch state, there will be a non-90 degree phase overrun or lag, so the phase error can also be used to detect and extract the frequency difference information.

In 2009, the Konkuk University’s (Seoul, Korea) research team proposed the matching control of MEMS vibration gyroscope by phase domain analysis. The control algorithm was to use the phase difference between the sense and drive modes as a reference for the mode mismatch. The measurement and control scheme is shown in Figure 13. This control method was applicable to the mode-matching of symmetric MEMS gyroscopes and could be widely used in optical sensors such as optical resonant mirrors and fiber laser accelerometers. It had good compatibility with devices in dual resonance mode [77], but a fatal flaw of this method was that the phase information was easily disturbed, so the mode-matching state could not be stably maintained.

Tuning based on phase information can be done not only based on the phase of the detected signal but also using the quadrature error signal. When the symmetric gyroscope is not mode-matched, a Coriolis effect is accompanied by quadrature error. The quadrature error signal is 90 degrees out of phase with the detection signal. Therefore, we can judge whether the working mode is matched or not, based on this error signal. In 2012, the National University of Defense Technology (Changsha, China) research team proposed changing the resonator’s stiffness coefficients at different azimuth angles by varying the DC voltage of the quadrature electrodes. The quadrature drift was used as the error feedback quantity of the quadrature loop to eliminate the quadrature error [78]. The digital control scheme of its FPGA is shown in Figure 14.

### 4.3. Amplitude Reference Combined with Phase Reference to Determine the Tuning Voltage

The combination of amplitude and phase information is bound to increase the credibility of the results. Based on the more mature two reference theories, some research teams combined phase and magnitude to determine the modal mismatch error, also known as CQC-PD (combining quadrature control and phase detection).

For tuning forks gyroscopes, the early operating mechanism was open-loop control. The open-loop control system has strong operability and high sensitivity, but the stability is poor and the system bandwidth is narrow. In particular, it cannot meet the requirements of the inertial navigation system for temperature stability. Therefore, the operation mechanism of force feedback is currently used for the tuning fork. The research team at Middle East Technical University in 2012 designed a controller to adjust the bandwidth parameter dynamically. The aim was to achieve a reasonable compromise between the two performance metrics. The bandwidth was adjusted by adjusting the closed-loop controller’s parameters that detected the modalities. The system block diagram is shown in Figure 15. Under the mode-matching condition, the system’s bandwidth was adjustable up to 50 Hz and the bias instability was 0.54°/h [79].

According to the conclusion that a second-order system’s amplitude-frequency and phase-frequency responses are evenly and oddly symmetric concerning the resonant frequency point, a real-time mode-matching technique was published by researchers at the Southeast University [80]. Researchers could determine the mismatch information by observing the response symmetry of the modulated signal. The test data demonstrated that the gyroscope achieved mode-matching while reducing the zero-offset stability from 5.89°/h to 1.26°/h and the angular random wander from 0.36°/√h to 0.079°/√h. The design of this scheme was based on an analog circuit and FPGA system, so it had a high signal transfer rate. The process of mode-matching could be done within 1 s.

The research team at Soochow University (Suzhou City, Jiangsu) also combined magnitude and phase to design a mode-matching algorithm for two-way control, as shown in Figure 16. The phase error of the quadrature response error was first extracted to change the tuning voltage to suppress the quadrature error. The other control loop was used to stabilize it in a micro-amplitude state. This method improved the accuracy of determining the tuning voltage by nearly 60 [81].

Conventional gyroscopes are single-mode driven, i.e., with only one drive signal and one sensitive signal. Due to the requirement of temperature stability, mode-matching generally requires the addition of temperature compensation. The researchers at Georgia Tech presented a dual-mode (DM) operation architecture on a flexible digital platform, which scheme actuated both modes of an axisymmetric gyroscope with two identical in-phase excitation signals. The parameters in the system could be tuned conveniently to optimize the performance. The DM operation reduced the bias drift of the gyros, leading to a significantly lower rate random walk (RRW). The performance repeatability and stability over the temperature of the DM gyro system were also improved considerably [82].

In 2018, the researchers proposed the dual-mode actuation architecture utilizing the difference of the individual mode outputs to cancel out the common-mode bias terms. The results could provide an inherently bias-free rate output with double sensitivity and SNR compared to the conventional SM. Furthermore, the quadrature-phase component of the output provided a mode-split indicator signal which could be used for in-run mode-matching. The dual-mode actuation scheme exhibited good temperature stability, with a 45× reduction in bias drift over a temperature range of 10–80 °C and over 100× reduction in the temperature drift of scale factor over 10–50 °C [83].

In 2021, researchers from Zhejiang University combined phase and amplitude to carry out modal matching. The linear tuning electrode is used for coarse adjustment and then fine adjustment. The gyroscope achieves an ARW of 0.011°/√*h* and BI of 0.229°/*h* under mode matching and with quadrature error suppressed by the quadrature nulling electrode.

The above control models are combined with amplitude and phase as the reference for modal matching. The control model of this kind of method is more complex than the first two kinds of electric tuning. But the adjustment accuracy is higher. The comparison of experimental data is shown in Table 4.

### 4.4. Summary

The above is a summary of some classic cases of mechanical tuning and electrical tuning. The former is more likely to break the accuracy bottleneck, but once the package does not achieve the desired effect, the subsequent increase in voltage tuning compensation is also unavoidable. Electrical tuning is indeed more flexible than mechanical tuning, so it is still the primary method of error suppression for symmetric gyroscopes, but to achieve higher accuracy and reliability, it is necessary to strive for excellence in the following links to meet high precision:The first item is modal frequency difference online real-time, non-perturbative detection. The existing frequency difference identification technology based on the detection mode injection signal will cause disturbance to the detection mode, because this technique needs to change the normal working state to extract the harmonic oscillator mode features, whether it is through phase information or amplitude information.The second item is constraints on injection signal modulation frequency and gyroscope measurement bandwidth. So that the Coriolis signal does not affect the identification of the frequency difference, the frequency of the injected signal needs to be higher than the signal to be measured; on the other hand, the modulation frequency also limits the measurement bandwidth of the gyroscope.The third item matches very high Q values with modal frequency difference tolerances. When the Q value of the harmonic oscillator increases sharply, the frequency response curve of the gyroscope is exceptionally steep. If the resonant frequency of the sensitive structure is 10 kHz and the Q is 500,000, its bandwidth is only 20 mHz. The extremely high Q value increases the isolation between operating modes, which requires high matching accuracy.The last item is matching modal frequency matching accuracy to drive frequency locking accuracy. Two aspects limit the frequency locking accuracy of the drive mode: the steady-state phase noise of the frequency control loop and the frequency tracking speed. To achieve highly reliable frequency matching, the phase noise and tracking rate of the drive frequency locking need to be optimized.

## 5. Emerging Algorithms Incorporated Modal Matching Technology

Various emerging algorithms are widely applied to control systems with the rise of cross-research between artificial intelligence and other disciplines.

In 2015, the research team at Peking University (Peking, China) designed a real-time mode-matching method for gyroscopes based on fuzzy control and neural network algorithms. The schematic block diagram of the algorithm is shown in Figure 17. Given the nonlinearity of the controlled object, a two-dimensional fuzzy controller is used to regulate the tuning voltage. A combined neural network algorithm is used for temperature error compensation. Since the driving frequency changes with the ambient temperature, it can be measured in the driving closed loop. Therefore, the modal frequency can be regarded as a feedback variable, and the tuning voltage can be changed to achieve modal matching control.

The three-layer BP (back propagation) neural network algorithm is used for control, as shown in Figure 18. The input, hidden, and output layers are M, q and L neurons. At the same time, the tansig function and pureline function are used as transfer functions of the hidden layer and output layer, respectively. The steepest descent method is used in the research process to adjust the weight matrix, $\omega_{ij}$, and $\omega_{ki}$ to minimize the training error. Then, the neural network controller can be used to predict according to the input. The mode-matching process can be achieved in less than 10 s.

The phase margin, gain margin, and sensitivity margin calculated from the experimental data reach the theoretical values. The system has sufficient stability and robustness in the temperature range of −40~80 °C. In addition, the gyroscope has a maximum bandwidth of 85 Hz and a bias instability of 5°/*h* under closed-loop control [35].

Later, the researchers at Hohai University (Nanjing, China) also used a fuzzy neural network algorithm to realize the stability control of the gyroscope. They innovatively proposed an adaptive fractal sliding mode control (SMC) scheme based on approximation. The scheme adopts a double loop recurrent fuzzy neural network (DLRFNN) to approximate the system uncertainty and disturbance. The simulation data shows that the model performs better in high precision and fast response [36,37,38].

## 6. Discussion

Given the symmetric MEMS gyroscope application field, its development direction will continue to break through various technical bottlenecks and combine more scientific and technological achievements to improve gyroscope accuracy. Based on the current research foundation, the following outlook is proposed to optimize symmetric gyroscope mode-matching technology in the future:Since most of the parameters of the gyroscope structure have temperature-sensitive characteristics, temperature stability is a significant challenge in ensuring accuracy. When the external environment temperature changes, the gyroscope structure parameters cannot change symmetrically, which causes a modal mismatch. Especially when facing extreme weather changes, accuracy is not guaranteed. Therefore, future mode-matching technology needs to consider the temperature compensation link. Curing the temperature compensation algorithm into the mode-matching algorithm can achieve multiple guarantees of accuracy.IC technology is a necessary technical basis for realizing various microsystems. The integration technology and microsystem manufacturing process are combined with mode-matching algorithms to form chip-level error suppression products. Such products can enhance system functions, and reduce size and power consumption. We can adapt them to the wide range of needs for available lightweight goods with high integration in various fields.With the consumer upgrades of today’s society, symmetric MEMS gyroscopes are no longer limited to high-end military fields. Civil and commercial areas such as autonomous driving, indoor guidance navigation, and medical and health care rely on inertial navigation. Therefore, it is vital to consider the people’s demand for intelligence and refinement. The mature combination and application of intelligent control algorithms with error suppression algorithms is the third important direction of mode-matching technology.

In summary, the mode-matching technique of symmetric MEMS gyroscope is mainly divided into two aspects. One is from the aspect of resonant oscillator structure through changing the structural parameters to achieve the symmetry of the two modal parameters. The other approach is to start with the peripheral control circuit by finding the system response in the presence of frequency errors with an algorithm. It provides some reference for applying symmetric MEMS gyroscope in the field of high precision.

## Figures and Tables

**Figure 1 micromachines-13-01255-f001:**
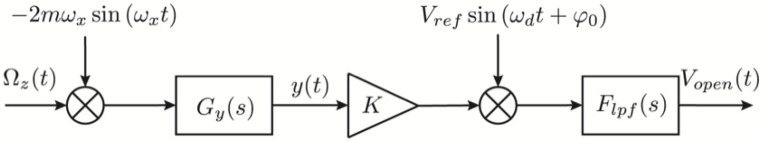
Schematic diagram of gyroscope open-loop detection.

**Figure 2 micromachines-13-01255-f002:**
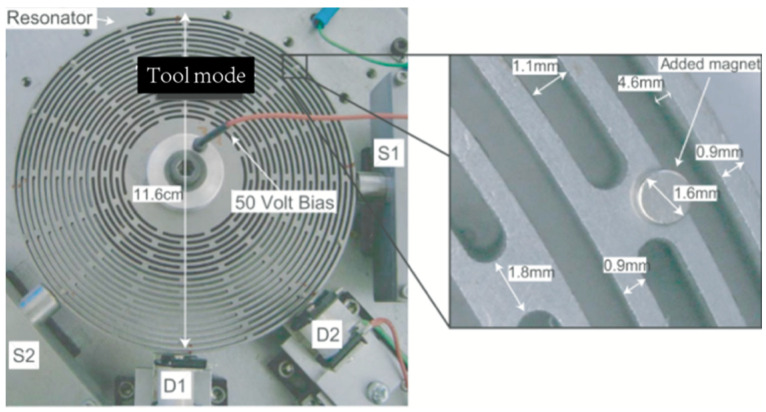
Schematic diagram of the mass perturbation process proposed by UCLA.

**Figure 3 micromachines-13-01255-f003:**
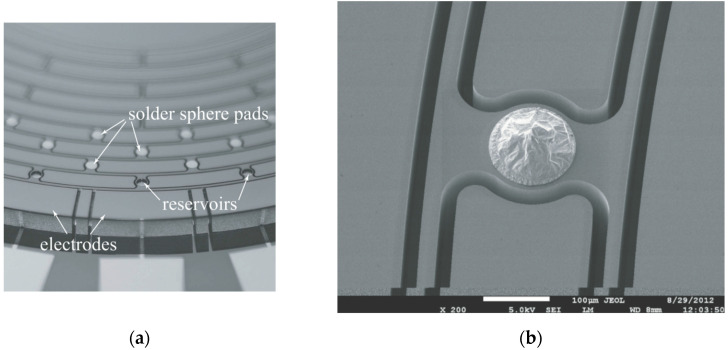
Schematic diagram of the structure of proposed mass perturbation tuning technology; (**a**) Structural diagram; (**b**) Reflow solder ball on the gold pad of the resonator.

**Figure 4 micromachines-13-01255-f004:**
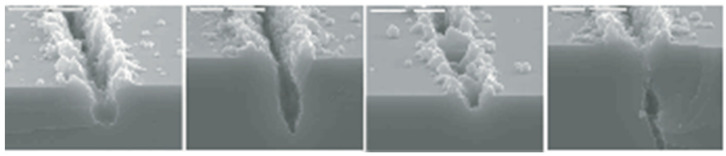
Example of resonant oscillator laser ablation profile from Newcastle University.

**Figure 5 micromachines-13-01255-f005:**
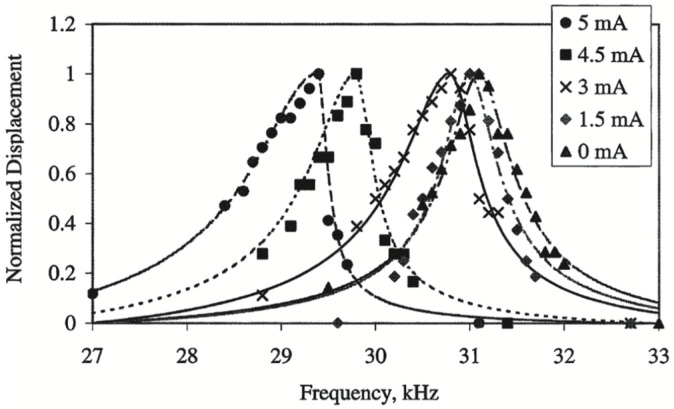
Simulation curve of local thermal stress effect correction proposed from UC Berkeley.

**Figure 6 micromachines-13-01255-f006:**
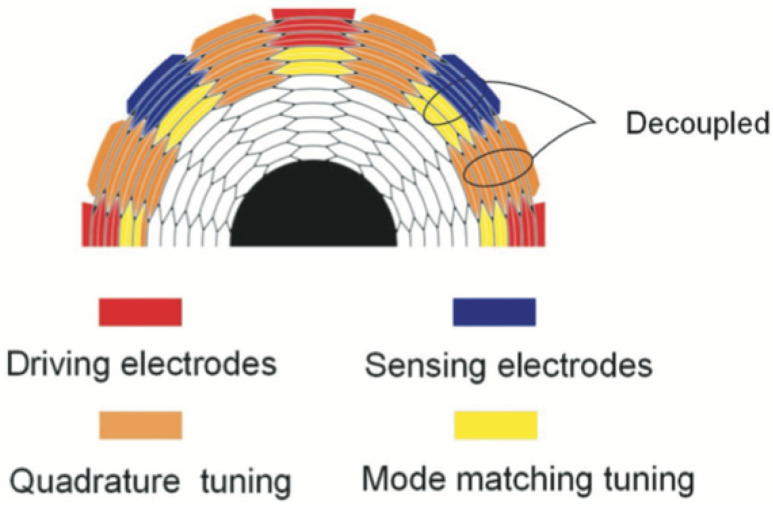
Honeycomb disc resonant gyro developed by the National University of Defense Technology.

**Figure 7 micromachines-13-01255-f007:**
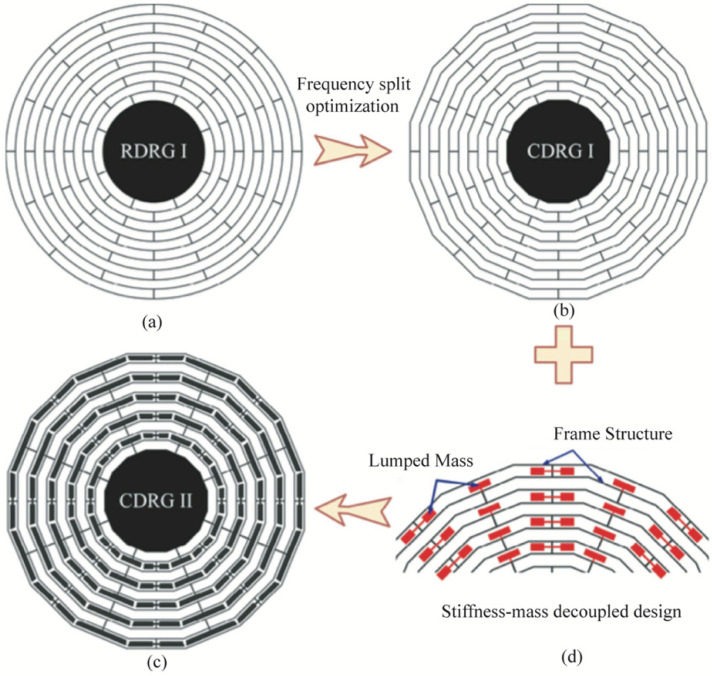
Evolution of a new spider-web type disc resonant gyro designed by Soochow University; (**a**) disk resonator gyroscope (RDRG I); (**b**) simple cobweb-like disk resonator gyroscope (CDRG I); (**c**) stiffness-mass decoupled design; (**d**) final design of cobweb-like disk resonator gyroscope (CDRG II).

**Figure 8 micromachines-13-01255-f008:**
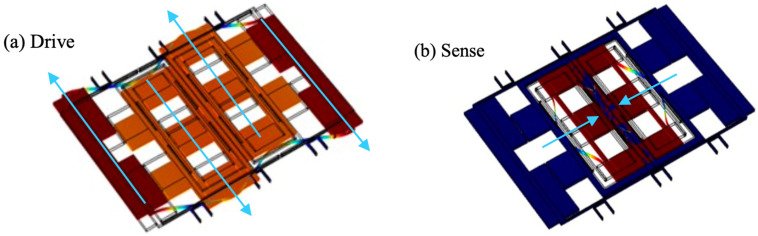
Drive and sense motions of the ladder gyroscope; (**a**) Working state of driving mode; (**b**) Working state of sense mode.

**Figure 9 micromachines-13-01255-f009:**
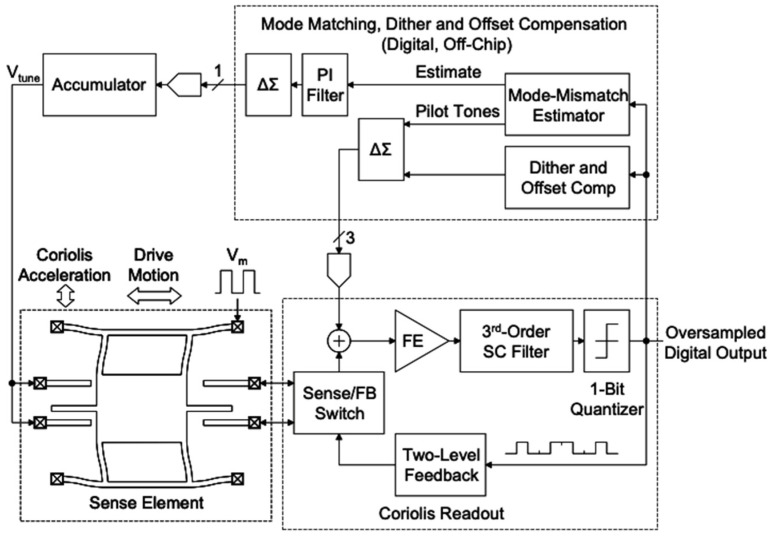
Gyroscope readout circuit designed.

**Figure 10 micromachines-13-01255-f010:**
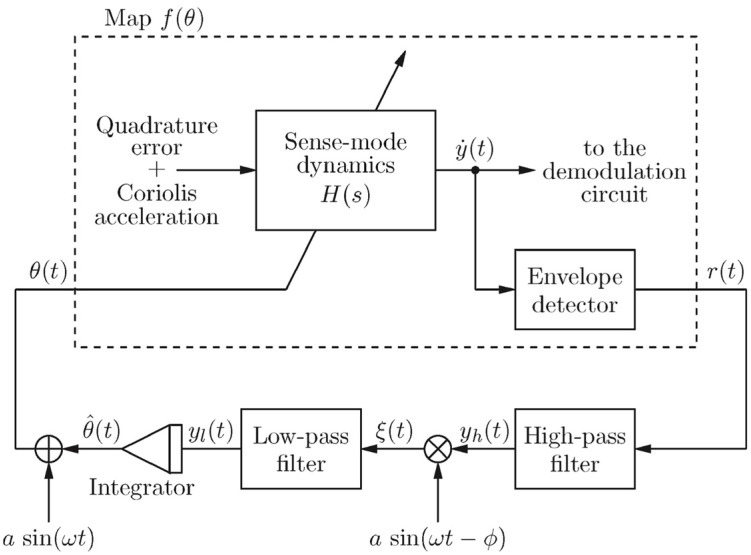
Block diagram of the proposed mode-matching control loop by Trento.

**Figure 11 micromachines-13-01255-f011:**
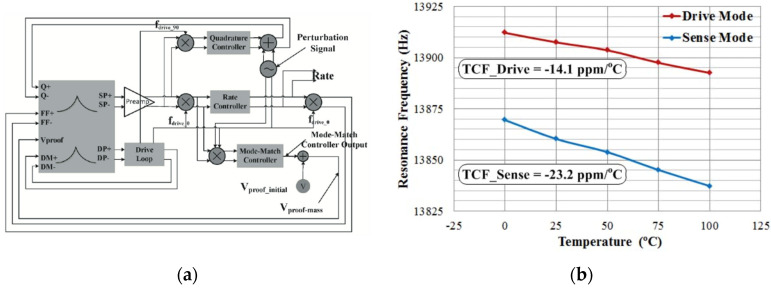
Auto-matching technique from the Middle East Technical University: (**a**) Block diagram of the closed-loop control system (**b**) Allen variance before and after modulation.

**Figure 12 micromachines-13-01255-f012:**
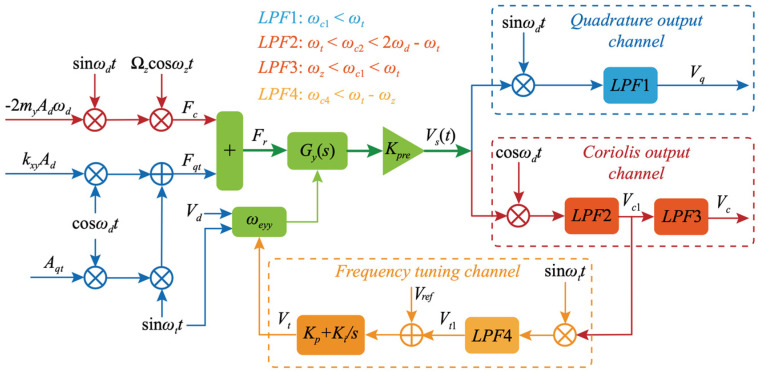
Control block diagram of the mode-matching algorithm based on quadrature modulation from Southeast University.

**Figure 13 micromachines-13-01255-f013:**
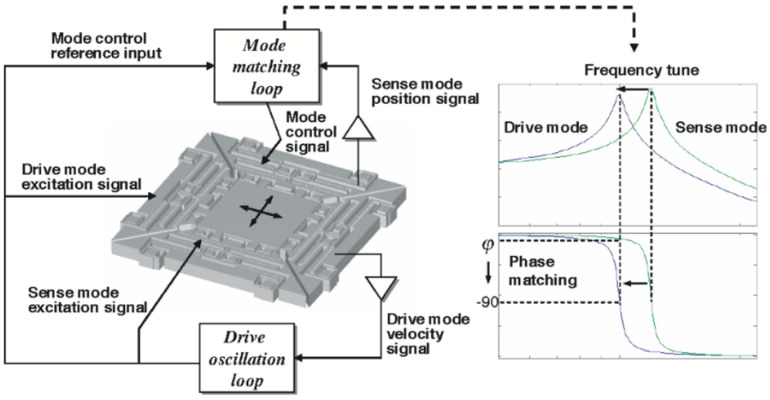
The mode-matching method based on phase domain analysis from the Konkuk University.

**Figure 14 micromachines-13-01255-f014:**
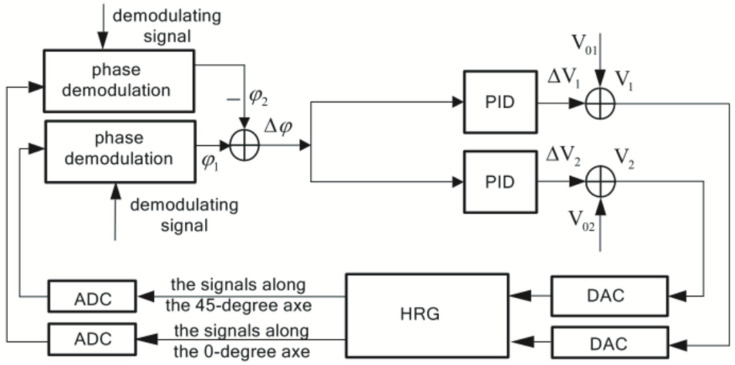
Block diagram of the orthogonal error from the National University of Defense Technology.

**Figure 15 micromachines-13-01255-f015:**
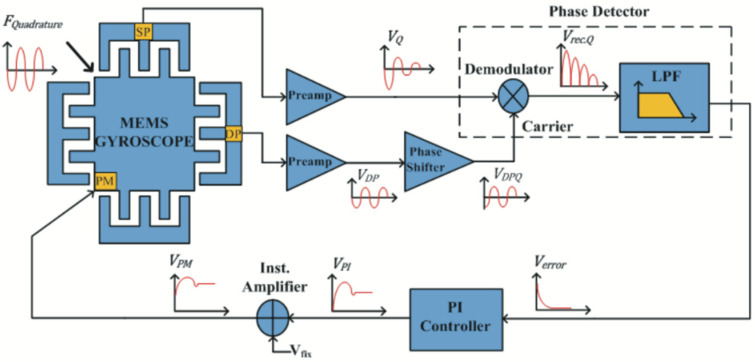
Block diagram of the mode-matching system from Middle East Technical University.

**Figure 16 micromachines-13-01255-f016:**
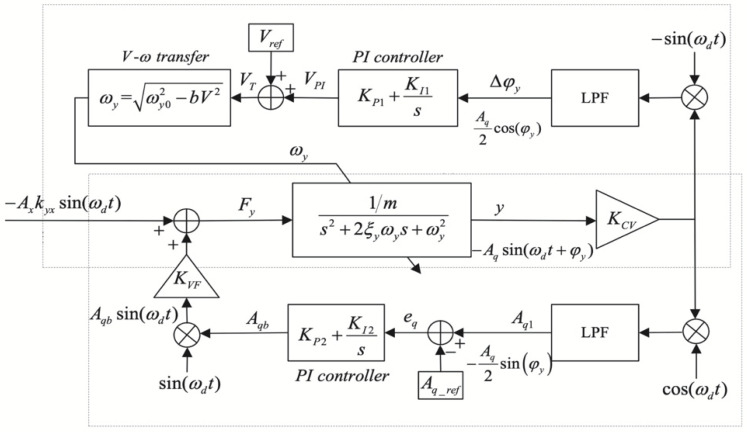
The framework of the mode-matching system with dual-phase and amplitude control from Soochow University.

**Figure 17 micromachines-13-01255-f017:**
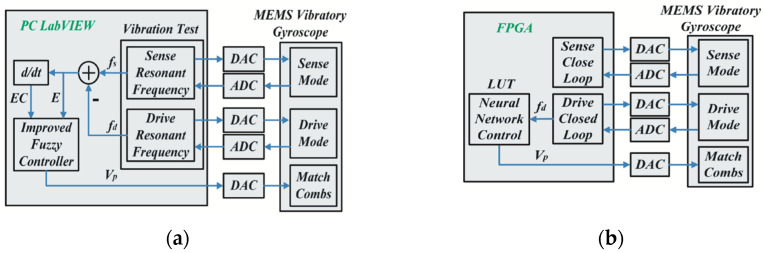
Mode-matching control method proposed by Peking University’s researchers: (**a**) block diagram of fuzzy control, and (**b**) block diagram of neural network algorithm.

**Figure 18 micromachines-13-01255-f018:**
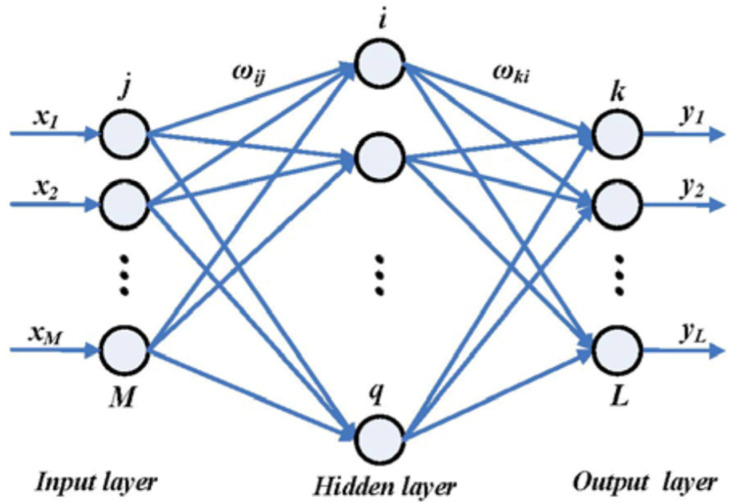
The architecture of a three-layer BP neural network controller.

**Table 1 micromachines-13-01255-t001:** Comparison of mode-matching schemes for structural mass augmentation and subtraction methods.

Design Unit	Technical Approach	Drive Frequency $f_{x}$	Frequency Difference after Tuning Δf	Others
Georgia Institute of Technology	Mass increment	23.5 kHz	≈200 Hz	Lower operability
UCLA	Mass increment	1.6 kHz	<1 Hz	High precision, susceptible toelectromagnetic interference
ATA Inc.	Mass increment	14 kHz	80 mHz	Fast response and high accuracy
University of Nottingham	Mass subtraction	35 kHz	<35 Hz	Only for trimming of single and two pairs of vibration patterns
Newcastle University	Mass subtraction	17 kHz	<0.1 Hz	High space utilization, more complex structure processing
National University of Defense Technology	Mass increment	23,170 Hz	0.4 Hz	The frequency split linearly related to the removed mass

**Table 2 micromachines-13-01255-t002:** Comparison of mode-matching schemes for the structural mass increase and decrease method.

Design Unit	Design Innovation	Drive Frequency $f_{x}$	Frequency Difference after Tuning Δf	Others
UC Berkeley	Thermistors as basic materials for sensitive structures	31 kHz	20 Hz	Low power consumption, poor temperature stability
National University of Defense Technology	Honeycomb disc resonance gyro	17 kHz	<0.1 Hz	High space utilization and excellent mechanical properties
Suzhou University	Spider web type disc resonance gyro	18 kHz	5.6 Hz	Static performance improvement
Murata Manufacturing and Kanazawa Murata Manufacturing	Ladder structure	9170 Hz	0.34 Hz	High precision, fast response, good temperature stability
Politecnico di Torino	A unique mechanical spring configuration	11,014 Hz	<1 Hz	Relatively low cost, high mechanical sensitivity

**Table 3 micromachines-13-01255-t003:** Comparison of electrostatic tuning schemes with amplitude as reference.

Design Unit	Types of Gyroscopes	Drive Frequency $f_{x}$	Frequency Difference after Tuning Δf	Others
UC Berkeley	Dual mass gyroscope	15 kHz	<50 Hz	Integrated, low noise and power consumption, not high accuracy.
University of Trento	Transverse pendulum gyroscope	4.5 kHz	<1 Hz	Easy to implement a system, high volume of collected data required
Middle East Technical University	Dual mass gyroscope	14.1 kHz	<40 Hz	ARW Improved and adjustable bandwidth, lower matching accuracy
Southeastern University	Dual mass gyroscope	3.9 kHz	0.3 Hz	Good temperature stability, need for external disturbance signal
Southeast University	Ring gyroscope	3923 Hz	0.3 Hz	No spectrum analysis required, low hardware resource requirements
Nanjing University of Science and Technology	Disk gyroscope	9746 Hz	<0.01 Hz	Suitable for DRG operating in the *n* = 3 wineglass mode

**Table 4 micromachines-13-01255-t004:** Comparison of electrostatic tuning schemes with dual reference to phase and amplitude.

Design Unit	Types of Gyroscopes	Drive Frequency $f_{x}$	Frequency Difference after Tuning Δf	Others
Southeastern University	Dual mass gyroscope	3780 Hz	<1 Hz	Easy to implement the system, high volume of collected data required
Suzhou University	Dual mass gyroscope	5550 Hz	<1 Hz	Good temperature stability, need for external disturbance signal
Middle East Technical University	Dual mass gyroscope	14 kHz	<1 Hz	ARW improved and adjustable bandwidth, lower matching accuracy
Southeastern University	Dual mass gyroscope	3780 Hz	<1 Hz	Easy to implement the system, high volume of collected data required
Georgia Institute of Technology	SD-BAW gyroscope	2.629 MHz	0.0062Hz	Shallow temperature drift, self-calibrating in terms of scale factor
Zhejiang University	Dual-mode decoupled single-mass micromechanical gyroscope	4979 Hz	0.15 Hz	Bias instability (BI) and angle random walk (ARW) below 0.1

## Data Availability

Not applicable.

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
