# Peer review of "A Review of Symmetric Silicon MEMS Gyroscope Mode-Matching Technologies"

_micromachines, 2022, doi:10.3390/mi13081255_

Round 1

Reviewer 1 Report

The manuscript is well written and organized.

However, the cited references are not adequate and up to date.

Please add relevant references, particularly since 2021.

Author Response

Thank you for your thorough review and insightful comments. We addressed all your comments and modified the manuscript as described below. The corresponding changes were highlighted in the manuscript.

Point :  The manuscript is well written and organized. However, the cited references are not adequate and up to date. Please add relevant references, particularly since 2021.

Response :  In this paper, the research results of citations [76, 79, 80 ] are achieved during 2021 to 2022. On this basis, I added citations [ 83-88]. The research results of these two papers were completed in 2021.

Reviewer 2 Report

The following issues can be considered to further improve this manuscript. 

1.      In page 20, Emerging algorithms incorporated modal matching technology,  a real-time mode-matching method for gyroscopes based on fuzzy control and neural network algorithms can be explained.

2.      What are the advantages of the proposed review paper?

3.      There are some applications of MEMS gyroscope  where your techniques could be of some interest .  such as  DOI 10.1109/TFUZZ.2021. 3094717,  DOI : 10.1109/TFUZZ.2021.3064704, DOI : 10.1109/TSMC.2020. 2979979 . A comment on these techniques and applications would be valuable.

Author Response

Thank you for your thorough review and insightful comments. We addressed all your comments and modified the manuscript as described below. The corresponding changes were highlighted in the manuscript.

Point 1:  In page 20, Emerging algorithms incorporated modal matching technology,  a real-time mode-matching method for gyroscopes based on fuzzy control and neural network algorithms can be explained.

Response 1:  

I revised it from  line 668 to line 674. Specific modifications are as follows:

The three-layer BP (back propagation) neural network algorithm is used for control, as shown in Figure 18. The input, hidden, and output layers are M, q and L neurons. At the same time, the tansig function and pureline function are used as transfer functions of the hidden layer and output layer, respectively. The steepest descent method is used in the research process to adjust the weight matrix  and to minimize the training error. Then, the neural network controller can be used to predict according to the input. The mode-matching process can be achieved in less than 10 seconds.

Point 2:   What are the advantages of the proposed review paper?

Response 2:

I revised it from line  125 to  line 131. Specific modifications are as follows:

Previous reviews of symmetrical gyroscopes are generally based on structure, materials and working principles. There is less in-depth analysis on error suppression technology. However, the accuracy of symmetrical gyroscope is an important parameter to reflect its performance. In order to understand the method of improving the accuracy of gyroscope, this paper classifies and summarizes some classical achievements. Based on the existing achievements, the future development direction is prospected. It provides theoretical basis for researchers in some related fields.

Point 3:  There are some applications of MEMS gyroscope  where your techniques could be of some interest .  such as  DOI 10.1109/TFUZZ.2021. 3094717,  DOI : 10.1109/TFUZZ.2021.3064704, DOI : 10.1109/TSMC.2020. 2979979. A comment on these techniques and applications would be valuable.

Response 3: The above three articles have been added to the citation list [84-86]. The text is modified as follows:

The researchers at Hohai University also used a fuzzy neural network algorithm to realize the stability control of the gyroscope. They innovatively proposed an adaptive fractal sliding mode control (SMC) scheme based on approximation. The scheme adopts a double loop recurrent fuzzy neural network (DLRFNN) to approximate the system uncertainty and disturbance. The simulation data shows that the model performs better in high precision and fast response [84-86].

Reviewer 3 Report

The review considers the designs of symmetrical MEMS gyroscopes, mechanical and electrical ways to reduce the frequency splitting. The manuscript contains quite a lot of references to recent works and may be useful to specialists working in the field of MEMS gyroscopes.

There are some remarks.

1. In section 2.2, the use of crystalline quartz, fused quartz and silicon is discussed. Here it should be noted significant differences in the properties of quartz crystal and fused quartz. Fused quartz has no anisotropy, has a very small coefficient of expansion (this leads to very low thermoelastic losses) and has a very low level of impurities (this leads to very low bulk internal friction). As a result, the use of fused quartz makes it possible to manufacture resonators with very high Q-factor.

2. Section 1 contains formulas (5)-(6) describing the displacements of the oscillator if the frequencies of the two modes are not equal. Sections 3 and 4 of the manuscript describe mechanical and electrical methods for reducing this frequency splitting. However, it is not clear from the text of the manuscript what magnitude of the frequency splitting MEMS should have. That is, the relationship between the MEMS error and the magnitude of the frequency splitting should be indicated.

3. On the page 3 (line 117) there is a typo: probably should be "100kHz" instead of "100k".

4.  DOI is specified only for some papers, this makes it difficult to find them.

5. It is necessary to improve the English language of the manuscript.

Author Response

(The authors gave the same response as above.)
